# Genome Profiling of H3k4me3 Histone Modification in Human Adipose Tissue during Obesity and Insulin Resistance

**DOI:** 10.3390/biomedicines9101363

**Published:** 2021-09-30

**Authors:** Daniel Castellano-Castillo, Bruno Ramos-Molina, Wilfredo Oliva-Olivera, Luis Ocaña-Wilhelmi, María Isabel Queipo-Ortuño, Fernando Cardona

**Affiliations:** 1Instituto de Investigación Biomédica de Málaga, Universidad de Málaga, 29010 Málaga, Spain; daniecas22@gmail.com (D.C.-C.); oliva_olivera@hotmail.com (W.O.-O.); 2Grupo de Obesidad y Metabolismo, Instituto Murciano de Investigación Biosanitaria (IMIB-Arrixaca), 30120 Murcia, Spain; brunoramosmolina@gmail.com; 3Unidad de Cirugía Metabólica, Hospital Clínico Virgen de la Victoria, 29010 Málaga, Spain; luisowilhelmi@hotmail.com; 4Unidad de Gestión Clínica Intercentros de Oncología Médica, Hospitales Universitarios Regional y Virgen de la Victoria, Instituto de Investigación Biomédica de Málaga (IBIMA)-CIMES-UMA, 29010 Málaga, Spain; 5UGC Pediatría, Instituto de Investigación Biomédica de Málaga, Hospital Regional de Málaga, 29010 Málaga, Spain; fernandocardonadiaz@gmail.com; 6Department of Surgical Specialties, Biochemistry and Immunology School of Medicine, University of Malaga, 29010 Málaga, Spain

**Keywords:** epigenetics, H3k4me3, adipose tissue, obesity, insulin resistance

## Abstract

Background: Adipose tissue (AT) dysfunction is involved in obesity-related comorbidities. Epigenetic alterations have been recently associated with AT deterioration in obesity conditions. In this work, we profiled the H3K4me3 histone mark in human AT, with special emphasis on the changes in the pattern of histone modification in obesity and insulin resistance (IR). Visceral AT (VAT) was collected and subjected to chromatin immunoprecipitation (ChIP) using anti-H3K4me3 antibody and then sequenced to obtain the H3K4me3 genome profile. Results: We found that most of the H3K4me3 enriched regions were located in gene promoters of pathways related to AT biology and function. H3K4me3 enrichment at gene promoters was strongly related to higher mRNA levels. Differentially expressed genes in AT of patients classified as non-obese, obese with low IR, and obese with high IR could be regulated by differentially enriched H3K4me3; these genes encoded for pathways that could in part explain AT functioning during obesity and insulin resistance (e.g., extracellular matrix organization, PPARG signaling or inflammation). Conclusions: In conclusion, we emphasize the importance of the epigenetic mark H3K4me3 in VAT dysfunction in obesity and IR. The understanding of such mechanisms could give rise to the development of new epigenetic-based pharmacological strategies to ameliorate obesity-related comorbidities.

## 1. Introduction

Adipose tissue (AT), traditionally seen as a mere storage tissue [1,2], is nowadays considered as an endocrine organ that can modulate systemic metabolism and energy metabolism in other organs [1,2,3]. 

A dysfunctional AT is believed to be in part responsible for the development of metabolic disorders associated with obesity, and it is thought to increase the risk for obesity-related diseases, such as diabetes or cancers [4,5]. Therefore, a better understanding of the molecular and environmental factors that can trigger AT deterioration could help to develop therapeutic approaches to mitigate some pathological conditions linked to obesity and related comorbidities. 

Epigenetic mechanisms have recently emerged as important factors involved in metabolic dysfunction in the onset of obesity [6]. In this regard, epigenetics is highly influenced by nutrition, and it has been suggested as a possible causal factor implicated in AT dysfunction in obesity conditions [7]. 

Histone modifications can be affected by key metabolic pathways [6,7] that are usually altered during obesity [8,9,10]. However, despite the central role that epigenetics might exert in AT biology, the chromatin profiling of histone marks in human visceral AT (VAT) has not been described so far. In this study, we have described for the first time the genomic profile of the lysine trimethylation of histone 3 (H3K4me3) in human VAT samples by using ChIP-seq technologies. In addition, we have established the H3K4me3 profile in human VAT samples from non-obese subjects and obese subjects with low and high insulin resistance (IR). 

## 2. Methods 

### 2.1. Sample Collection

AT biopsies were from patients recruited for either bariatric surgery or cholecystectomy surgery at Virgen de la Victoria University Hospital (Málaga, Spain) and stored at −80 °C until analysis. 

Clinical and anthropometric data of the selected patients (*n* = 4 per group) are shown in Appendix A. A control group including non-obese participants (BMI < 30 mg/kg^2^; NOb) and two groups including subjects with obesity (BMI > 30 mg/kg^2^) and with either low (HOMA-IR < 4; Ob.LIR) or high IR (HOMA-IR > 9; Ob.HIR) were included (Figure 1A).

All participants gave their informed written consent, and the study was reviewed and approved by the Ethics and Research Committee of the Virgen de la Victoria University Hospital.

### 2.2. Chromatin Immunoprecipitation Sequencing (ChIP-Seq)

Chromatin immunoprecipitation followed by next generation sequencing (ChIP-seq) was performed as previously described [11]. Briefly, 100 mg of VAT was cross-linked after what nuclei were purified. Then, nuclei were lysed, and the chromatin was sheared by sonication using a Bioruptor UCD-300 (Diagenode, Liège, Belgium). A total of 5 μg of chromatin were used for immunoprecipitation, which were performed by incubating the chromatin with 1.5 μg of anti-H3K4m3 (ab8580, abcam) fixed to protein G-coated magnetic beads (Dynabeads Protein G; Thermo Fisher Scientific, Carlsbad, CA, USA) at 4 °C overnight. Chromatin was released and purified (MinElute PCR Purification Kit, Qiagen, Hilden, Germany) and DNA samples were sequenced using Hiseq4000 (Illumina, San Diego, CA, USA) and paired-end mode (2 × 100 bp).

### 2.3. RNA Sequencing (RNA-Seq)

Total RNA was extracted using an RNEasy Lipid Tissue Mini Kit (Qiagen, Crawley, UK) following the manufacture’s protocol. The total RNA was converted into cDNA using the Transcriptor High Fidelity cDNA Synthesis kit (Roche, Penzberg, Germany). The library construction was performed using the Accel-NGS^®^ 1S Plus DNA Library Kit (Swift Biosciences, Ann Arbor, MI, USA) and were sequenced using a HiSeq 4000 instrument (Illumina, San Diego, CA, USA) and paired-end mode (2 × 100 bp).

### 2.4. ChIP-Seq Bioinformatic Processing and Analysis

Trimming and alignment were performed using the software QuasR (git clone https://git.bioconductor.org/packages/QuasR) and using human genome hg38. To generate robust peaks, bam files were merged prior to peak calling using the mergeBamByFactor function in SystemPipeR (git clone https://git.bioconductor.org/packages/systemPipeR). Peak calling was performed using MACS2 (Model-based Analysis of ChIP-seq) (http://github.com/taoliu/MACS/BSD/BSD) software (narrow mode and *q*-value threshold < 1 × 10^−9^) and the peaks were annotated using ChIPseeker. 

Differential analysis was performed using the bdgdiff function in MACS2. For this analysis, Y chromosome was omitted to avoid gender biases. 

Annotated peaks were analyzed for gene set enrichment analysis (GSEA) with the software clusterProfiler (git clone https://git.bioconductor.org/packages/clusterProfile) for Reactome, GO and KEGG terms for the 3000 top genes ranked by the peak *q*-value. For known motif enrichment analysis, we used Homer (Hypergeometric Optimization of Motif EnRichment) (configureHomer.pl v4.11 (10-24-2019) [GPLv3]). The analysis of the data was performed using R Studio.

### 2.5. RNA-Seq Bioinformatic Processing and Analysis

Alignment was performed against human genome hg38, and the counts over feature were analyzed using Subread (subread-align and featureCounts functions, respectively) (git clone https://git.bioconductor.org/packages/Rsubread). Differential gene expression analysis was performed using DESeq2 (git clone https://git.bioconductor.org/packages/DESeq2), while GSEA was performed as previously described. Integration and final analysis of the RNA-seq and ChIP-seq data sets was performed in R Studio (http://www.rstudio.com/products/rstudio/download/) (RStudio, Boston, MA, USA). 

## 3. Results

### 3.1. A Whole Picture of H3K4me3 Profile in Human Adipose Tissue

H3K4me3 is mainly present in the promoter of active genes [12]. Accordingly, we found that in human AT the H3K4me3 histone mark was enriched at gene promoters, with nearly 75% of all the peaks (Figure 1B). Almost 17,500 genes had H3K4me3 enrichment in their promoter (flank distance of 1000 kb) (Figure 1C). Furthermore, there was an enrichment of reads and peaks found near the transcriptional start site (TSS) (Figure 1C,D). The motif enrichment analysis of such peaks showed that H3K4me3 genomic location matched with transcription factor binding sites (TFBS) of important regulators of AT biology such as *PPARG* (PPARE), *TFAP2C* (*AP2-gamma*), *Isl1* and *RXR*, among others (Figure 1E). To have a wider panorama we performed a GSEA of the top genes (3000 top peaks ranked by q-value). The complete list of pathways found by Reactome, GO and KEGG terms can be found in Appendix A (Sheet 1). Overall, we found that genes enriched with the H3K4me3 mark were related to pathways involved in AT function and biology (Appendix A).

### 3.2. H3K4me3 Mark Is Associated with Gene Expression in Human Adipose Tissue

H3K4me3 is a mark related to gene activation, and genes with such modification in the promoters are expected to be more expressed [13]. Therefore, we next performed RNA sequencing (RNAseq) in order to find out whether the H3K4me3 profile in human AT was related to activation of gene expression (data store similarity depicted in Appendix A). Our analysis revealed that out of the 18,933 active genes, 13,435 (more than 70%) had H3K4me3 enrichment in their promoter (Figure 2A). These H3K4me3-enriched genes showed on average a higher mRNA expression (Figure 2B). 

Usually, the level of enrichment of a histone modification at certain regions is defined by both the width of the peak and the number of reads found at Summit (Pile Up) [13]. Therefore, we compared the width and the “Pile Up” of genes expressed and non-expressed in human AT and found that expressed genes had higher enrichment of H3K4me3 marks defined by the “Pile Up”, while their peaks were more restricted in width (Figure 2C,D). Non-expressed and H3K4me3-enriched genes had weaker enrichment and the modification was sparse in the promoter. For those genes that were expressed and marked with the histone modification in their promoters, there were significant positive correlations between the average gene expression with both, the “Pile Up” and the width of the peaks (Figure 2E,F). 

We next analyzed the pathways enriched for genes that are expressed (active genes) and marked with H3K4me3 in their promoters (Figure 2G). A complete list of pathways is depicted in Appendix A. Overall, we found that pathways implicated in AT function and development were enriched: signaling by receptor tyrosine kinases, autophagy, WNT signaling, mTOR signaling, AMPK signaling, TNF signaling pathway, FOXO signaling and regulation of cholesterol by SREBP. 

### 3.3. Analysis of H3K4me3 Enrichment in AT during Obesity and IR

In order to understand the relationship between histone marks in AT with either obesity and IR we analyzed the H3K4me3 profile in AT from NOb, Ob.LIR and Ob.HIR subjects. To increase the power analysis and find out common patterns according to the status of obesity and/or IR, we merged the BAM files in all the groups and called peaks using MACS2 peak caller (Narrow mode, *q*-value < 1 × 10^−9^) [13]. Comparisons between conditions showed high data store similarities though the participants mostly clustered according to their BMI and HOMA-IR in PCoA (Appendix A). 

### 3.4. Differential Analysis of AT H3K4me3 Profile in Obesity and IR

To search for differentially regulated genes between groups, we first performed differential expression analysis using DESeq2. The complete comparisons between groups are depicted in Appendix A. We found 1304 genes differentially expressed between Ob.LIR and NOb subjects, 2829 between Ob.HIR and NOb subjects and 2946 between Ob.HIR and Ob.LIR (*p* < 0.05). 

Next, we aimed to analyze whether differentially expressed genes could be related to H3K4me3 enrichment at their promoters between study groups. To perform that, we compared H3K4me3 enrichment among groups. The integrative analysis revealed that 664, 755 and 1085 genes were differentially expressed and enriched in their promoter with H3K4me3 when Ob.LIR vs. NOb, Ob.HIR vs. NOb, and Ob.HIR vs. Ob.LIR groups were compared, respectively (Figure 3A–C and Appendix A). 

These differentially expressed and H3K4me3 enriched genes were then subjected to GSEA. We found that genes differentially expressed and enriched with the H3K4me3 mark in the Ob.LIR group with respect to the NOb group were included in pathways related to the immune system, protein processing and location in endoplasmic reticulum or energy metabolism (Figure 3A). For genes differentially expressed and enriched with H3K4me3 in their promoters in the comparison between Ob.HIR vs. NOb groups, we found enrichments for pathways related to apoptosis, cell fibrosis, hypoxia or vesicular transport among others (Figure 3B). When Ob.HIR and Ob.LIR groups were compared, we found that the genes differentially expressed and enriched in H3K4me3 were included in pathways involved in protein processing, mitochondria processes, energy generation, response to endoplasmic reticulum stress or Golgi function among others (Figure 3C). 

### 3.5. Changes in H3K4me3 Enrichment Are Related to Transcriptional Changes 

In order to understand whether the differences in mRNA expression between study groups were associated with differences in the enrichment of H3K4me3 marks we further analyzed the direction (more or less enriched) of the differentially expressed and H3K4me3-enriched genes (Figure 4A–C). We then calculated which was the mean mRNA Fold-change (FC) of the genes with H3K4me3 upregulated, downregulated and with peaks with double direction (upregulated and downregulated) for the comparisons between groups. Interestingly, we found that the genes that were enriched with H3K4me3 marks at their promoters had a positive FC in the intergroup comparison, while genes with a reduction in the H3K4me3 mark had decreased mRNA levels (Figure 4D). For the genes that had at their promoter both increase and decrease of H3K4me3, we observed a positive FC, although this was weaker than the observed for genes with only upregulation of H3K4me3 (Figure 4D).

### 3.6. Top 10 Differentially Expressed and H3K4me3 Enriched Genes and Their Relationship with Metabolic Variables

Then we analyzed the relationship of the 10 top (5 upregulated and 5 downregulated) concordant (same FC direction for both, mRNA expression levels and H3K4me3 enrichment) genes with several metabolic variables of the study subjects (Table 1). Regarding the H3K4me3-upregulated genes, we observed a positive correlation between waist and HOMA-IR with several genes such as *ADRA2A*, *ED1*, *LCP1*, *PCSK6*, *BAG3*, *FAM198B*, *DOK3*, *NANOS1* and *TUBB2B* (Figure 4E). Furthermore, BMI was positively associated with *EDN1*, while glucose was negatively associated with *ZBTB16* and positively correlated to *NMNAT2*. Cho and LDL-Cho presented a negative association with the expression of *PCSK6* (Figure 4E).

On the other hand, the genes with downregulation of H3K4me3 at their promoters were mainly associated with waist, HOMA-IR and insulin. Moreover, several genes were negatively related to BMI as *GRIA1*, *GRIK3*, *ISL1* and *CYP2G1P*. TG levels were negatively correlated with *NRXN3* and *CYP2G1P*, while Cho and LDL-Cho correlated in a positive way with *ANKRD21* (Figure 4F).

## 4. Discussion

In this work, we have described for the first time (to the best of our knowledge) the signature of the H3k4me3 mark (a histone modification associated with positive gene expression) in human VAT from obese patients with high and low levels of IR, as well as, from non-obese controls. We found that H3K4me3 is enriched at gene promoters and is related to gene expression levels in human AT. The genes enriched with this mark at their promoters are genes related to key pathways implicated in AT biology and function. In addition, we observed that the differential enrichment of the H3K4me3 mark is highly associated with the mRNA levels of different genes related to AT biology in the onset of obesity and IR. 

H3K4me3 is a well-known histone mark related to gene expression. In our analysis, we observed that the H3K4me3 mark in human AT is enriched at the TSS, which is in line with the reported data present in the literature [14]. Motif enrichment analysis showed that genomic regions with H3K4me3 modification were enriched for binding sequences of known adipose regulators such as PPARG known to be the key factor in adipogenesis [15,16] and required for adipocyte identity and survival of mature adipocytes (loss of PPARG activity is related to the acquisition of a fibroblast-like phenotype in adipocytes during obesity); EBF1 (a direct target of *PPARG*) and EBF2 are transcription factors necessary for adipogenesis of 3T3-L1 cells [17]. In mice, EBF1 downregulation has been related to increased lipolysis, adipose hypertrophy and reduced insulin sensitivity [18]; TFAP2C (AP2-gamma) together with TFAP2A have been proposed to be key regulators of WNT signaling-dependent lipid droplet biogenesis, or ISL1, which seemed to regulate early adipogenesis by inhibiting *PPARG* expression. Therefore, H3K4me3 peaks are enriched for important motif sequences that are known to regulate AT development and function.

We have also shown that genes enriched for H3K4me3 at their promoter have higher mRNA expression levels. This is in accordance with the fact that H3K4me3 is a well-established active mark of gene expression. However, not all the genes with H3K4me3 promoter enrichment are expressed (approximately 13% of the peaks), and 29% of the expressed genes do not present H3K4me3 enrichment. These results indicate that in human AT, the presence of H3K4me3 mark is neither necessary nor sufficient for the expression of certain genes as previously described for other metabolic organs such as the liver [13]. Our results, however, strongly suggest that the genes with this modification in their promoter present higher expression levels compared with genes without this mark. In addition, we showed that the peak architecture could be related to gene expression level as well. Thus, in our study, mRNA expression levels were positively correlated with the height (“PileUp”) and the width of the peaks as it has already been shown in mouse liver [13]. In addition, we found that genes marked with H3K4me3 at their promoters were associated with important pathways involved in AT function and involved in obesity and IR (e.g., TCA cycle and respiratory electron transport, metabolism of carbohydrates, autophagy, mitochondrial processes, energy generation processes, oxidative phosphorylation or non-alcoholic fatty liver disease (NAFLD) among others). Altogether, our data suggest that in human AT, the histone modification H3K4me3 could stimulate gene expression and that genes enriched with H3K4me3 at their promoters mostly encode for key factors in AT development and function.

Further analysis revealed that the enrichment of the H3K4me3 mark is highly dependent of the presence of obesity and/or IR. We found that a high proportion of the differentially expressed genes also had differentially enriched H3K4me3 mark at their promoters (50%, 26% 36% for the comparisons Ob.LIR vs. NOb, Ob.HIR vs. NOb and Ob.HIR vs. Ob.LIR were respectively). GSEA of differentially expressed and H3K4me3 enriched genes showed that this histone modification could be related to the expression of genes related to AT function, suggesting a new epigenetic mechanism involved in the pathophysiology of AT at the onset of obesity and/or IR. For instance, in the comparison Ob.LIR vs. NOb individuals, we observed an enrichment of pathways related to immune cells, extracellular matrix organization, the TCA, or mitochondrial organization and respiration, all of them known to be altered during obesity [19,20,21,22,23]. Notably, the signaling by nuclear receptors (LXRA or LXRB (NR1H2 or NR1H3), ESR or PPARG signaling pathways) was also enriched, pathways that can regulate AT development and lipid metabolism [15,24]. Another enriched pathway was the SLIT/ROBO pathway, which is also involved in AT biology [25]. Interestingly, a member of the SLIT/ROBO family, SLIT2, has been shown to be secreted by beige fat and promote thermogenesis, energy expenditure and, therefore, to improve glucose homeostasis in vivo [26]. 

For the comparison between O.HIR vs. NOb groups, we found again genes related to immune cells and more interesting metabolic pathways as lipid regulation or integration of energy metabolism, which comprises regulation of insulin or ChREBP regulatory factors, among others. ChREBP (carbohydrate response element binding protein) is a carbohydrate sensor that is important in AT de novo lipogenesis by modulating key metabolic genes (pyruvate kinase, *FAS*, Acetyl-CoA carboxylase, GP-acyl transferase) and of which de-regulation is known to alter insulin sensitivity [27]. 

Additionally, when the O.HIR vs. O.LIR groups were compared to find out the possible influence of IR in the profile of the histone modification H3K4me3 in AT, we found that the genes differentially expressed and enriched for the H3K4me3 mark were included in pathways involved in Golgi vesicle transport, ER protein location, mitochondria structure, location and function, IL-6 pathway, the TCA cycle and GLUT4 transport to the cell membrane. AT inflammation is known to be related to a low-grade systemic inflammation that contributes to IR, being IL-6 an important mediator of such status [28], while GLUT4 is the major insulin-stimulated glucose transport, which is decreased during IR [29]. Dysfunctional AT is related to altered mitochondria function (essential for lipid metabolism, energy production and adipocyte biology) [23], and the alteration of which leads to lipid accumulation, generation of ROS species and dysregulation of the adipokine profile delivered by AT, contributing to IR [23]. 

We further investigated the genes differentially expressed and enriched in H3K4me3 and described that 87.5% of the differentially regulated peaks were enriched in Ob.LIR compared to NOb, 74% were enriched in Ob.HIR compared to NOb, while most of the peaks (91.5%) were enriched in Ob.LIR compared to Ob.HIR. Remarkably, we previously demonstrated by H3K4me3 ChIP followed by qPCR that several genes related to AT biology and function displayed a higher H3K4me3 enrichment in the promoter in obese subjects with low IR compared to the rest of the study participants [12]. These results could suggest that H3K4me3 methylation and de-methylation processes could be impaired during obesity and/or IR in human AT [12]. In addition, given the high association between H3K4me3 enrichment and mRNA expression levels, our analysis might suggest that changes in H3K4me3 marks induced by obesity and/or IR could be an important epigenetic factor involved in dysregulation of AT in both metabolic conditions.

We then focused on the top downregulated and upregulated genes in which we observed genes usually related to AT biology. As examples of genes that could be H3K4me3 regulated with a role in obesity and IR: (1) *CXCL14* is a chemokine that in mice has been shown to be released by brown adipocytes and that exerts a beneficial role in white adipose tissue causing increased of insulin sensitivity in obese mice [30]. Therefore, higher levels of this cytokine in Ob.LIR could be in part responsible for the low IR level shown compared to Ob.HIR; (2) *ZBTB16*, which has been proposed as a regulator of adipogenesis in epigenetic studies, has been shown to stimulate adipogenesis and induce brown-like adipocyte formation in bovine primary cells [31]; (3) another upregulated gene was *ROCK2*, an inhibitor of adipogenesis in 3T3L1 cells, while partial *ROCK* KO mice display a pronounced thermogenic rewiring of white AT, AT mass gain and an amelioration of IR [32,33]; (4) *PLIN2*, which is located in the droplet surface, can inhibit glucose uptake in several models including differentiated 3T3L1 cell lines [34,35]; therefore, the higher regulation of *PLIN2* in Ob.HIR would also be in concordance; (5) *CD36* is upregulated both in high IR states. Lack of *CD36* in a *Cd36*-KO mouse model presented lower adiposity with alteration in leptin production and higher insulin sensitivity [36]. In a more recent study, CD36 has been described to enhance AT inflammation and cell death in diet-induced obese mice while *CD36*-KO mice showed improved insulin sensitivity [37], while its blockage in 3T3L1 has been described to enhance insulin signaling [38]. Therefore, H3K4me3-mediated *CD36* overexpression in Ob.HIR compared to Ob.LIR could be contributing to IR in these subjects; (6) *F13A1* is also upregulated in Ob.HIR, whose absence induces in diet-induced mice “healthy obesity” [39]; (7) lack of *DDRGK1* has been shown to cause ER stress (present in obesity) [40] and stress-induced apoptosis in cancer cells and hematopoietic cell lines [41]. Therefore, the H3K4me3-induced expression of *DDRGK1* in Ob.LIR subjects could be regarded as a protective factor compared to Ob.HIR subjects. 

## 5. Conclusions

As conclusions, we have demonstrated that in human VAT, H3K4me3 is located at the promoter of genes involved in AT development and function. H3K4me3 could regulate gene expression in human VAT, and likewise, genes enriched for H3K4me3 at their promoters displayed higher mRNA levels. Furthermore, H3K4me3 enrichment was highly dependent on the presence of obesity and/or IR, which in turn was associated with changes in mRNA levels. These changes have been shown to impact AT function during obesity and IR, therefore being key factors in the etiology of both conditions. This study provides new evidence on the epigenetics of human VAT during obesity and metabolic disease, which could reveal new clues for the development of new epigenetic-based tools or pharmacological treatments against obesity and obesity-related disorders. 

## Figures and Tables

**Figure 1 biomedicines-09-01363-f001:**
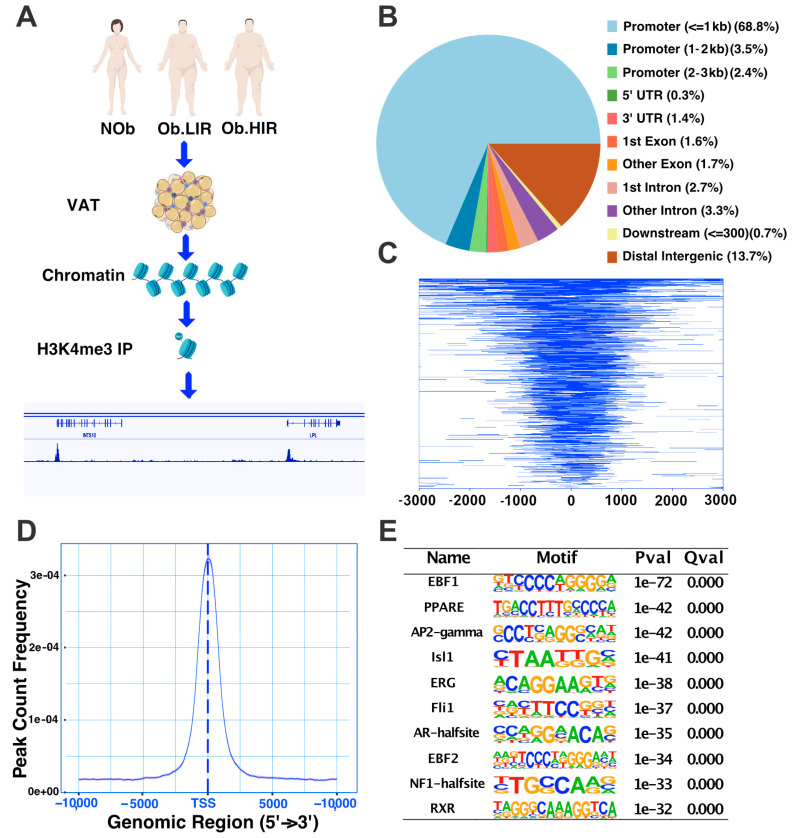
(**A**) Schematic workflow. In this study, VATsamples were collected from non-obese individuals (Nob) and obese subjects with either low and high insulin resistance (O.LIR and O.HIR, respectively). The samples were processed to extract the chromatin, which was immunoprecipitated (IP) using magnetic beads coated with H3K4me3 antibodies. The DNA from the IP chromatin was then purified and sequenced in order to obtain the H3K4me3 histone modification profile over the genome. Annotation of the H3K4me3 peaks found showed that most of the enrichment, nearly 75%, was located inside promoter regions (**B**). Enrichment feature distribution also showed a wide gene enrichment of H3K4me3 in adipose tissue located mainly between 1Kb up and downstream of transcription start sites (TSS) of the genes (**C**). As is shown in (**D**), most of the reads sequenced were concentrated around the TSS. (**E**) Top of enriched known motives present in H3K4me3 peaks in human AT.

**Figure 2 biomedicines-09-01363-f002:**
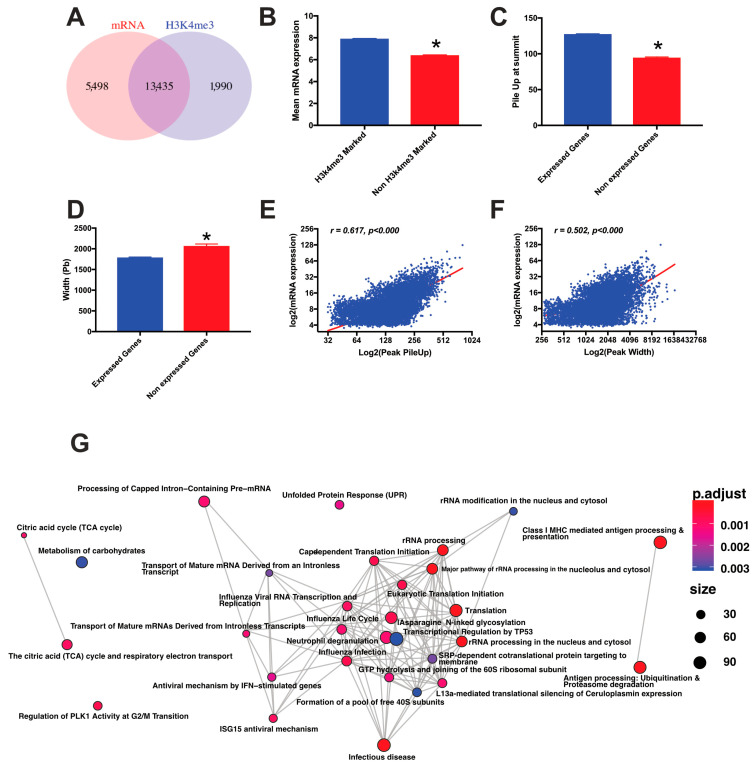
Integration of H3k4me enrichment with the RNA seq. The intersection between the genes enriched at their promoter with H3K4me3 and active are displayed in (**A**); (**B**) average gene expression level of genes with and without enrichment of H3K4me3 at their promoter regions. (**C**) Pile Up (reads at the summit of the peaks) average for expressed and non-expressed genes; (**D**) width of the peak for expressed and non-expressed genes; (**E**) and correlation between the Pile Up and the gene expression level; (**F**) correlation between the peak width and the average mRNA levels; (**G**) set of genes that were enriched for H3K4me3 at their promoter and expressed for 3000 Top genes ranked by peak *p*-value (Reactome). * means *p* value < 0.05.

**Figure 3 biomedicines-09-01363-f003:**
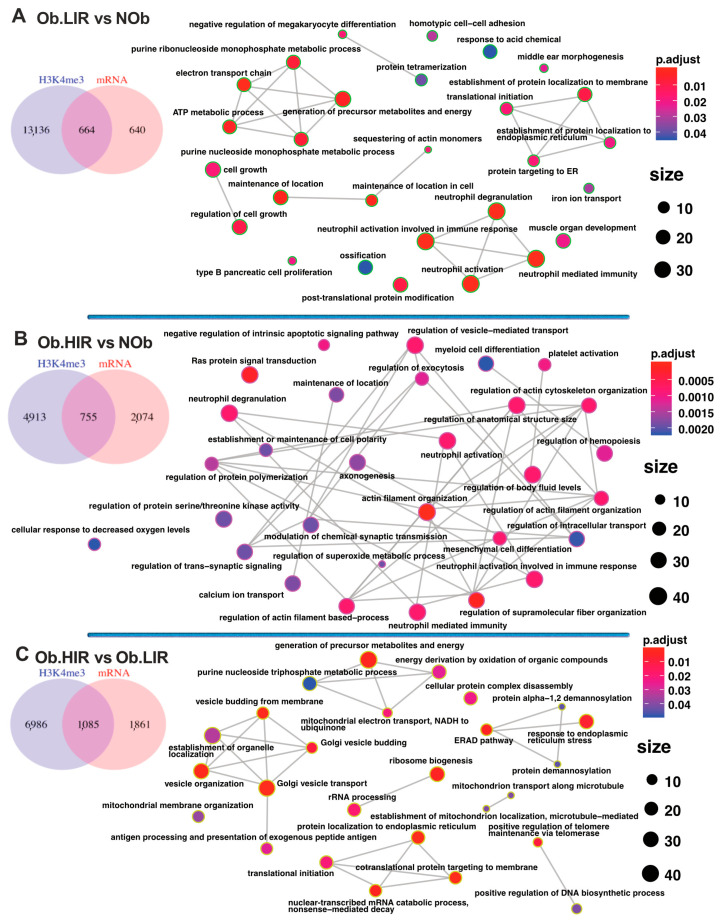
Intersection of genes differentially expressed (mRNA) and enriched for H3K4me3 and enriched pathway analysis for the comparison: Ob.LIR vs. NOb (**A**), Ob.HIR vs. NOb (**B**) and Ob.HIR vs. Ob.LIR (**C**). Pathways analysis for GO terms. The green, pink and yellow-green rings represent pathways enriched for the comparisons Ob.LIR vs. NOb, Ob.HIR vs. NOb and Ob.HIR vs. Ob.LIR, respectively.

**Figure 4 biomedicines-09-01363-f004:**
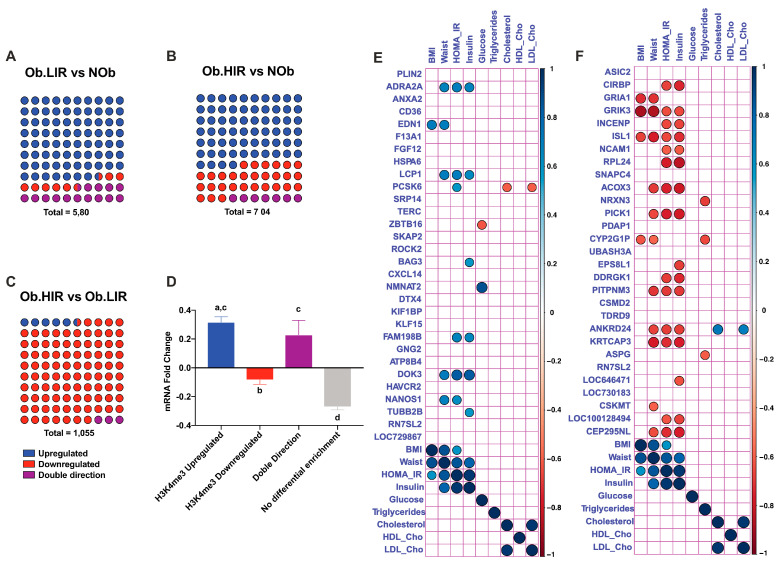
Percentage of genes upregulated, downregulated, with bidirectional enrichment of H3k4me3 and correlation between top 10 upregulated and downregulated genes. Genes, both with differential mRNA and H3k4me3 enrichment were subtracted, and the direction of the enrichment was calculated for the different comparisons between groups: Ob.LIR vs. NOb (**A**), Ob.HIR vs. NOb (**B**) and Ob.HIR vs. Ob.LIR (**C**). Mean Fold-change (FC) mRNA for the different genes according to H3K4me3 enrichment direction, and the subset of genes in which differential H3k4me3 enrichment was not present (**D**). Correlations between upregulated and downregulated genes and clinical variables (**E**,**F**). H3K4me3 enrichment direction is shown by color blue for genes enriched; red for genes decreased; and blue for genes with both, enriched and decreased peaks. In the correlation analysis, colored genes mean *p* < 0.05 according to Spearman correlation. Blue dots mean positive correlations, while red dots mean negative correlations.

**Table 1 biomedicines-09-01363-t001:** Top 5 concordant genes (between mRNA and H3K4m3 changes) among study groups. In blue and cream, genes with enriched and decreased H3K4me3 mark, respectively, for each comparison between groups.

Ob.LIR vs. NOb			Ob.HIR vs. NOb			Ob.HIR vs. 0b.LIR		
Gene	FC	*p*	Gene	FC	*p*	Gene	FC	*p*
*CXCL14*	1.75	0.000	*ADRA2A*	1.59	0.000	*CD36*	1.62	0.000
*TERC*	2.33	0.000	*LCP1*	1.79	0.000	*F13A1*	0.89	0.004
*ZBTB16*	2.09	0.000	*NANOS1*	1.27	0.000	*DTX4*	0.89	0.004
*EDN1*	1.28	0.001	*ROCK2*	0.77	0.000	*ATP8B4*	0.97	0.007
*KLF15*	1.42	0.001	*PLIN2*	1.55	0.000	*FGF12*	1.12	0.014
*GRIA1*	−1.63	0.000	*PICK1*	−1.10	0.000	*DDRGK1*	−0.90	0.000
*GRIK3*	−2.50	0.000	*ACOX3*	−0.77	0.000	*PDAP1*	−0.98	0.000
*NRXN3*	−1.82	0.013	*GRIK3*	−2.76	0.000	*RPL24*	−1.02	0.000
*CSMD2*	−2.24	0.015	*PITPNM3*	−1.54	0.000	*INCENP*	−0.96	0.000
*ISL1*	−0.89	0.016	*EPS8L1*	−2.05	0.000	*CSKMT*	−3.09	0.000

## Data Availability

All data generated or analysed during this study are included in this published article (and its Appendix A).

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
