# Peer review of "Genome Profiling of H3k4me3 Histone Modification in Human Adipose Tissue during Obesity and Insulin Resistance"

_biomedicines, 2021, doi:10.3390/biomedicines9101363_

Round 1

Reviewer 1 Report

The study by Daniel Castellano-Castillo and colleagues investigating epigenetic histone methylation mark (H3K4me3) is of high relevance for the potential readers. 

While the introduction could be regarded as particularly short for some readers, I admit that it is clear and possess indispensable information.

Another issue which could be controversial is the number of subjects per group ( N=4), however, as I scientist who collected VAT samples I am aware of methodological issues associated with this type of material and the fact that VAT is not the most easily collected material. Another not an easy task was to find obese subjects with low HOMA-IR. 

The most critical issue about patients is the sex, particularly the lack of men in obeseLIR group. However, as stated earlier, it could be too time-consuming task to collect patients with the same gender or both genders and keep the obese low / high HOMA-IR status. 

My question is associated with obese high HOMA-IR subjects - taking into account their biochemical data some of them could suffer from T2DM. This information along with drugs affecting lipid profile should be included. 

The results are interesting and nicely described. The discussion is really long, however, it is also complexed and easy to follow. Manuscript also possesses many methodological details.

Some test editing should be performed. Examples : 

  1. Section 3.2 : Our analysis revealed that out  the 18,933 active genes 
  2. Section 3.4 To performed that,
  3. We found that H3K4me3 enriched at gene promoters and is related to gene expression levels in human AT
  4. which seemed to regulate early adipogenesis by inhibiting PAPRG expression.
  5. For the comparison between O.HIR vs. NOb groups we found again genes related to immune cells and more interesting metabolic pathways lipid regulation or integration 325 of energy metabolism,
  6. ChREBP (carbohydrate response element binding protein) is a carbohydrate sensor that that is important in AT de novo lipogenesis by modulating key 328 metabolic genes

I recommend the paper to be published after minor text editing and congratulate authors on obtaning interesting and novel results of the epigenetics of VAT upon metabolic imbalance.

Author Response

The study by Daniel Castellano-Castillo and colleagues investigating epigenetic histone methylation mark (H3K4me3) is of high relevance for the potential readers. 

While the introduction could be regarded as particularly short for some readers, I admit that it is clear and possess indispensable information.

Another issue which could be controversial is the number of subjects per group (N=4), however, as I scientist who collected VAT samples I am aware of methodological issues associated with this type of material and the fact that VAT is not the most easily collected material. Another not an easy task was to find obese subjects with low HOMA-IR. 

Response:We appreciate the comment. As pointed out, collecting human and apply some methodological approaches to adipose tissue can be very challenging. This is especially true for chromatin immunoprecipitation where is needed a sufficient amount of chromatin to get an appropriate enrichment of the modification. Also, ChIP sequencing requires of a large number of reads in order to have a good quality data store. In any case, we have used a similar number of samples per group as the observed in other studies also using adipose tissue though in a different biological context (Garcia-Eguren et Al. J Clin Endocrinol Metab. 2021). Also, we have performed the analysis by merging the bam files per group to get strong and trustworthy peaks per condition as indicated in other epigenetic studies (Dai et Al. Nat Commun. 2018) and therefore trying to minimize the effect of the small sample size.

The most critical issue about patients is the sex, particularly the lack of men in obeseLIR group. However, as stated earlier, it could be too time-consuming task to collect patients with the same gender or both genders and keep the obese low / high HOMA-IR status. 

Response:Thank you for the comment. Effectively the lack of men in obese LIR group is a weak point in our study. As the reviewer commented, collecting samples from obese subjects with low insulin resistance is challenging, and this is even more pronounced for the fact that we are very limited for the amount of tissue needed to perform both, a proper chromatin immunoprecipitation and whole RNA sequencing. Though gender could have an effect in the onset and pathophysiology of obesity and insulin resistance, recent studies have shown that obese women does not differ in their transcriptional profile from obese men (Rey et Al. Int J Mol Sci. 2021). In this study, they only found 51 genes differentially expressed in subcutaneous adipose tissue. Therefore, and due to the fact that the transcriptional profile should be in accordance with the epigenetic landscape (as shown in our work), we still believe that our work can be of big interest among the scientific community since show for the first time what is the profile of H3K4me3 in adipose tissue in the context of obesity and insulin resistance, and how the deregulation of this epigenetic modification might be related to the pathophysiology of such disorders.

My question is associated with obese high HOMA-IR subjects - taking into account their biochemical data some of them could suffer from T2DM. This information along with drugs affecting lipid profile should be included. 

Response:Thank you for the comment. Only one patient in the Ob.HIR group was diagnosed as diabetic and had glucose and cholesterol lowering drugs. We have introduced this information in the biochemical and anthropometrical table S1.

The results are interesting and nicely described. The discussion is really long, however, it is also complexed and easy to follow. Manuscript also possesses many methodological details.

Some test editing should be performed. Examples: 

  1. Section 3.2: Our analysis revealed that out  the 18,933 active genes 
  2. Section 3.4 To performed that,
  3. We found that H3K4me3 enriched at gene promoters and is related to gene expression levels in human AT
  4. which seemed to regulate early adipogenesis by inhibiting PAPRG expression.
  5. For the comparison between O.HIR vs. NOb groups we found again genes related to immune cells and more interesting metabolic pathways lipid regulation or integration 325 of energy metabolism,
  6. ChREBP (carbohydrate response element binding protein) is a carbohydrate sensor that that is important in AT de novo lipogenesis by modulating key 328 metabolic genes

Response:We thank the reviewer for pointing out these text typos, we have corrected the text.

Reviewer 2 Report

This study is interesting but presents a methodological problem. The number of patients in the groups is very low (n = 4). In addition, the M / F distribution is very different.The authors should increase the number of patients.

The authors should improve the quality of the figures. 

Author Response

This study is interesting but presents a methodological problem. The number of patients in the groups is very low (n = 4). In addition, the M / F distribution is very different. The authors should increase the number of patients.

Response:We thank the reviewer for the comment. Collecting human adipose tissue sample can be too time-consuming task, especially for obese subjects with low insulin resistance. Besides, this is also affected for the fact that women undergo bariatric surgery more frequently than men, with some papers describing 80% of the total (Fuchs et Al. J Laparoendosc Adv Surg Tech A. 2015). For the number of patients per group, we have used a similar number of samples per group as the observed in other studies also using adipose tissue for ChIP sequencing (Garcia-Eguren et Al. J Clin Endocrinol Metab. 2021). Also, in order to minimize the effect of size we have performed the analysis by integrating the data sets per group to obtain a robust peak calling in which weak peaks or noise is removed (Dai et Al. Nat Commun. 2018). Regarding the M/F distribution effectively the lack of men in obese LIR group is a weak point in our study. This is due to the fact that we are very limited in the sample size (since chromatin immunoprecipitation and RNA sequencings require of a minimum of starting sample size) and as stated before in getting obese that fulfill the criteria of low IR. Though gender could have an effect in the onset and pathophysiology of obesity and insulin resistance, recent studies have shown that obese women does not differ in their transcriptional profile from obese men (Rey et Al. Int J Mol Sci. 2021), with only 51 genes differentially expressed between both obese groups. Therefore, and due to the fact that the transcriptional profile should be in accordance with the epigenetic landscape (as shown in our work), we still believe that our work can be of big interest among the scientific community since show for the first time what is the profile of H3K4me3 in adipose tissue in the context of obesity and insulin resistance, and how the deregulation of this epigenetic modification might be related to the pathophysiology of such disorders.

The authors should improve the quality of the figures. 

Response:We appreciate the reviewers comment. We have made changes in the figures to increase the visibility of the path names and help to clarify the content. We hope the changes performed fulfill with the criteria of the journal and of course we are willing to perform further changes if we are inquired.

Round 2

Reviewer 2 Report

the authors addressed my comments